# Exploring Epigallocatechin-3-Gallate Autoxidation Products: Specific Incubation Times Required for Emergence of Anti-Amyloid Properties

**DOI:** 10.3390/antiox11101887

**Published:** 2022-09-23

**Authors:** Mantas Ziaunys, Vytautas Smirnovas

**Affiliations:** Institute of Biotechnology, Life Sciences Center, Vilnius University, LT-10257 Vilnius, Lithuania

**Keywords:** amyloid, protein aggregation, antioxidant, epigallocatechin-3-gallate, aggregation inhibition

## Abstract

Amyloidogenic protein/peptide aggregation into fibrillar aggregates is associated with multiple amyloidoses, including widespread neurodegenerative disorders. Despite years of research and a well-understood mechanism, there are still very few treatments available for the increasing number of amyloid-related disorders. In recent years, the search for potential anti-aggregation compounds has shifted toward naturally occurring molecules, with one of the most promising being epigallocatechin-3-gallate (EGCG). This polyphenolic compound was shown to inhibit the aggregation of several amyloidogenic proteins/peptides, including amyloid-beta (related to Alzheimer’s disease) and alpha-synuclein (related to Parkinson’s disease). However, multiple reports have indicated its limited stability under physiological conditions and the possibility of EGCG autoxidation products being the actual inhibitory compounds. In this work, we explore how different EGCG autoxidation products associate with non-aggregated insulin, as well as how they affect its aggregation and resulting fibril structure. We also show that there is a specific incubation time required for the emergence of compounds, which alters the amyloid aggregation process.

## 1. Introduction

Protein aggregation into amyloid fibrils is linked with the onset and progression of multiple amyloidoses [1], including neurodegenerative Alzheimer’s or Parkinson’s diseases [2,3]. The process of such aggregate formation is still not fully understood, with new insights into the complex mechanism of fibril formation being reported each year [4,5,6]. Despite this progress, there are still very few treatment modalities or cures available [7,8], which makes it difficult to halt the ever-increasing number of reported amyloid-related disorders [9,10]. This makes it important to acquire a better understanding of not only the aggregation process, but also the wide variety of potential anti-amyloid compounds and their interactions with amyloidogenic proteins.

The search for potent, disease-altering compounds has encompassed a range of molecules of diverse size and structure [11]. This includes both small molecular weight compounds, such as gallic acid [12] or resveratrol [13], as well as monoclonal antibodies [14] and complex inhibitor mixtures [11]. In recent years, significant attention has been devoted to natural anti-amyloid compounds [15], including multiple polyphenols, flavones, and flavonoids [16,17,18]. One especially interesting compound, which was found to inhibit the aggregation of not only model amyloidogenic proteins (insulin, lysozyme) [19,20], but also neurodegenerative-disease-related proteins/peptides (alpha-synuclein and amyloid-beta) [21,22], was epigallocatechin-3-gallate (EGCG) [23]. This polyphenol has an abundant presence in green tea leaves [24] and has shown potent anti-amyloid activity at near-physiological conditions [21,25,26]. Coupled with its antioxidant activity [27], this makes EGCG a potential candidate for the treatment of amyloid disorders.

Alongside all the praise and positive reports that EGCG has received over the years, there have been several studies demonstrating various aspects of the compound, which may affect its application in altering the process of amyloid formation and its use in vivo. It has been shown that EGCG can easily undergo autoxidation at near-physiological conditions, with multiple different compounds forming after as little as two hours of incubation [28]. It has also been reported that the anti-amyloid effect is highly dependent on the environmental conditions, where a change in solution pH value, ionic strength, or sample agitation negated its ability to slow down protein aggregation [22,26,29]. Other observations include covalent modification of proteins after autoxidation [30], protein precipitation [31], as well as possible cytotoxic effects in vivo [32]. Conversely, it has also been shown that EGCG may not actually be the molecule that inhibits amyloid fibril formation and only its autoxidation products have this property [20,26].

In this work, we analyzed how EGCG autoxidation products affected the formation of insulin amyloid fibrils. For this study, a range of different autoxidation time EGCG samples were prepared and analyzed by absorbance spectroscopy. It was then examined how these samples affected non-aggregated insulin precipitation, the lag time of insulin fibrillization, the secondary structure of formed aggregates, and their ThT-binding properties. We show that EGCG requires a specific incubation time at the selected conditions in order to gain its aggregation-altering properties. We also demonstrate that the most potent components of the inhibitor mixture may be the large molecular size polymeric species.

## 2. Materials and Methods

### 2.1. Epigallocatechin-3-Gallate (EGCG) Autoxidation

EGCG (Fluorochem, Hadfield, United Kingdom, cat. No. M01719) was combined with 100 mL of 100 mM potassium phosphate (pH 7.4) buffer to a final EGCG concentration of 10 mM and mixed with a magnetic stirrer at 4 °C for 30 min (the buffer solution was cooled down to 4 °C prior to mixing). The resulting solution was then filtered using a Millipore Stericup Sterile Vacuum Filtration unit (150 mL capacity, 0.45 µm pore size), which was kept on ice during the entire filtration procedure. Afterward, the EGCG solution was distributed to 1.5 mL Eppendorf Safe-Lock test tubes (final volume was 1.5 mL), which were additionally sealed with parafilm to prevent evaporation. The test tubes were placed in a 60 °C dry-bath incubator for the selected timeframes (Table 1). For each condition, three samples were incubated and then combined into one 4.5 mL solution, which was immediately frozen at −20 °C. All procedures not related to incubation were performed at low temperatures (4 °C or on ice) to prevent any additional/unwanted EGCG autoxidation.

### 2.2. Aggregation Kinetics

Human recombinant insulin powder (Sigma-Aldrich, St. Louis, MO, USA, cat. No. 91077C) was dissolved in a 20% acetic acid solution, containing 100 mM NaCl (henceforth referred to as AC solution) to a final protein concentration of 400 µM (ε_280_ = 6335 M^−1^cm^−1^). Amyloidophilic dye thioflavin-T (ThT) was dissolved in MilliQ H_2_O, filtered through a 0.22 µm pore size syringe filter, and its stock solution concentration was set to 10 mM (ε_412_ = 23250 M^−1^cm^−1^). The previously prepared EGCG solution samples were thawed at 4 °C and an aliquot from each was diluted 10 times using the AC solution to a final concentration of 1 mM. For aggregation experiments, the insulin, ThT, and EGCG solutions were mixed to a final protein concentration of 200 µM, 100 µM ThT, and 25 µM EGCG (EGCG concentration was chosen based on a previous study [22]). For control samples, either non-incubated EGCG or 100 mM potassium phosphate (pH 7.4) buffer solutions were used (Table 1).

The prepared solutions were then distributed to 96-well non-binding plates (100 µL in each well, 4 repeats for every condition), using alternating sample placement similar to [22] in order to minimize the effect of temperature differences in the plate. Aggregation kinetics were monitored using a ClarioStar Plus plate reader by scanning ThT fluorescence emission intensity every 5 min (440 nm excitation and 480 nm emission wavelengths) at 60 °C without agitation. The reaction apparent rate constant, lag time, and end-point fluorescence intensities were determined as shown in Appendix B, Figure A1. All data processing and fitting were done using Origin (OriginLab Corporation, Northampton, MA, USA) software.

To examine the effect of EGCG samples on an alternative insulin fibrillization pathway, different aggregation conditions were used, based on a previous study [33]. Insulin powder was dissolved in a 100 mM sodium phosphate buffer (pH 2.4), containing 100 mM NaCl. Under these conditions and a final protein concentration of 200 µM, non-aggregated insulin exists in a dimeric state [34] and forms fibrils with a different secondary structure [33]. All other aggregation procedures were performed as in the previously described aggregation assay.

### 2.3. Fourier-Transform Infrared (FTIR) Spectroscopy

For FTIR spectroscopy, aggregated insulin samples were recovered from the 96-well plate used in the aggregation kinetic assay and combined to final volumes of 400 µL (from all four repeats). The samples were then centrifuged at 10,000 RPM for 15 min, after which the supernatant was removed and the fibril pellets were resuspended into 200 µL D_2_O, containing 400 mM NaCl. The addition of NaCl improves sedimentation and the replacement of H_2_O with D_2_O removes the H_2_O-specific band in the Amide I/I’ region of the FTIR spectra [35]. This centrifugation and resuspension procedure was repeated four times, after which the fibril pellet was resuspended into 50 µL of D_2_O with 400 mM NaCl. The resulting sample FTIR spectra were scanned using a Bruker Invenio S FTIR spectrometer, as described previously [36]. D_2_O and water vapor spectra were subtracted from the sample spectra, which were then baseline corrected and normalized between 1700 and 1580 cm^−1^. Half-height bandwidth (HHBW) [37] values were then determined for all spectra. All data processing was done using GRAMS (Thermo Fisher Scientific, Waltham, MA, USA) software.

### 2.4. Atomic Force Microscopy (AFM)

After the aggregation reaction, samples 1 and 24 were diluted 20 times using the AC solution to a final protein concentration of 10 µM. AFM images of the samples were obtained as described previously [36]. The AFM images were then analyzed using Gwyddion 2.57 software. Aggregate cross-sectional heights and widths were determined by tracing lines perpendicular to the fibril axes. The resulting height and width values (n = 30) were analyzed using an ANOVA one-way Bonferroni means comparison (*p* < 0.01).

### 2.5. UV/Vis Spectroscopy

Sample optical densities were measured using a Shimadzu UV-1800 spectrophotometer at 800 nm in a 3 mm pathlength cuvette. The OD_800_ of 100 mM potassium phosphate (pH 7.4) buffer was subtracted from the OD_800_ value of each sample. For each sample, three measurements were taken and averaged. For sample absorbance measurements, EGCG samples were thawed at 4 °C, diluted 100 times using the aforementioned potassium phosphate buffer to a final concentration of 100 µM, and distributed to UV-clear 96-well plates (200 µL in each well, 5.2 nm pathlength). Absorbance spectra were measured in a ClarioStar Plus plate reader at room temperature from 220 to 800 nm. The absorbance spectra of the control sample without EGCG were subtracted from all sample spectra, which were then normalized based on the absorbance value at 800 nm. For each condition, three measurements were taken and averaged.

For insulin precipitation measurements, samples were prepared similarly as described in the aggregation kinetics Section 2 (without the addition of ThT) to a final insulin concentration of 200 µM and a range of EGCG concentrations. The samples were then mixed, and held at room temperature for 15 min, after which their OD_800_ was measured as described previously. For each condition, three measurements were taken and averaged.

To determine the level of competitive binding between ThT and EGCG to insulin fibrils, aggregated insulin samples (200 µM) and ThT, EGCG stock solutions (0 h and 80 h incubation samples) were combined to yield solutions containing 100 µM insulin fibrils, 50 µM ThT, and 50 µM EGCG. As controls, the same concentrations of ThT or EGCG were combined with fibrils separately. The resulting solutions (1 mL each) were mixed vigorously for 10 s, incubated for 30 min at room temperature, and centrifuged at 10,000 RPM for 15 min. Afterward, the supernatants were carefully removed (800 µL) and placed in UV-clear 96-well plates (200 µL in each well, 4 repeats for every condition). The sample absorbance measurements were performed as described previously. In order to compare the resulting absorbance spectra, the control ThT and EGCG sample spectra (when they were combined with fibrils separately) were added together.

### 2.6. EGCG Sample Concentration

The prepared EGCG samples were thawed at 4 °C, placed in 0.5 mL maximum volume 10 kDa concentrators (400 µL solution), and centrifuged at 5000 RPM for 10 min. This procedure was repeated 4 times with a total of 1.6 mL solution and the concentrates/permeates were separated. The distribution of concentrate and permeate was determined by weighing each combined solution using analytical scales, assuming similar solution densities. In some cases, the combined concentrate volume remained relatively high, and an additional round of centrifugation was carried out. The concentrate solution volumes were then made equal for all conditions by diluting them using their respective permeate solutions to one-third of the total initial sample volume (533 µL). This was done to resuspend the high molecular weight compounds in a 3 times lower volume (increase their concentration 3 times relative to the initial solutions). The concentrate and permeate solutions were then used in the aggregation kinetic, as well as UV/Vis spectroscopy assays, as described previously.

## 3. Results

An array of EGCG samples were incubated for different amounts of time at 60 °C, ranging from 10 min to 80 h. The incubation times were selected with exponentially increasing intervals (10 min intervals at the beginning, 10 h intervals towards the end). A visual observation of the resulting samples revealed a gradual color change from very faint pink at short incubation times to dark brown after long incubation periods (Appendix B, Figure A2C). Analysis of the absorbance spectra of the initial and final samples (after dilution to 100 µM) revealed multiple differences (Figure 1A). The peak position at 275 nm shifted towards 264 nm and the band at ~330 nm became less clearly expressed. There was also an increase in the absorbance values between 400 and 550 nm, which was previously shown to be caused by EGCG autoxidation products [28]. Since the clearest distinction between the incubated and non-incubated samples was the shift in the absorbance peak at 275 nm, this position was traced for all EGCG samples (Figure 1B). This peak position did not experience any changes up to 1 h of incubation, after which it began gradually moving towards 271 nm at 20 h. After this point, there was a considerable jump in peak position to 267 nm at 25 h and further movement towards 264 nm at 70–80 h. Based on this data, it appears that 1 h of incubation is necessary for significant changes to begin occurring and that there is also a similar critical point between the 20 and 25 h mark.

After a few hours of incubation, the samples started becoming opaque, suggesting the formation of precipitates. After even longer time periods, small amounts of dark precipitates began forming and collecting at the bottom of the test tubes. In order to separate the insoluble particles from the rest of the solutions, the samples were centrifuged in 10 kDa concentrators, and the absorbance spectra of concentrates and permeates were measured, as described in the Section 2. In the case of the non-incubated sample, both the concentrate and permeate had nearly identical spectra to the control, meaning that no particles were separated (Figure 1C). Interestingly, increasing incubation times caused the volume of permeate to gradually decrease, which may indicate the formation of particles that prevent the membrane’s permeability (Figure 1D). This reduction in permeate volume continued up to the 30–40 h incubation samples, after which the permeability began to suddenly increase, with the 80-h incubation sample having similar concentrate/permeate volumes as the control. Such an increase can be explained by the large number of smaller pore-obstructing particles (present after 30–40 h of incubation) forming fewer massive structures and allowing the solution to easily permeate the concentrator membrane.

Absorbance spectra of the 8-h sample concentrate and permeate were significantly different from the original sample. The concentrate absorbance spectrum was roughly twice as intense over the entire wavelength range, while the permeate had lower absorbance and a reduction in the peak at 227 nm. This suggests that large or insoluble particles were separated from incubated sample solutions. Examining the concentrate absorbance values at 270 nm revealed that there was a discontinuity at the 5 h mark, after which the absorbance value change followed a different trendline. The initial gradual increase up to this point can be attributed to the overall spectral changes occurring in EGCG samples at 270 nm and was not due to an increase in large particles, separated by the 10 kDa concentrator (Appendix B, Figure A2A).

The different incubation time EGCG samples, as well as their concentrates and permeates, were examined in an insulin aggregation assay. Surprisingly, none of the samples up to numbers 12–14 (5–9 h) displayed any aggregation inhibiting properties (Figure 2A). The same was true when using both the permeates (Figure 2B) and concentrates (Figure 2C). After this time-point, the aggregation lag time increased 3.5 times when using the original samples, 2.5 times with permeates, and 4.5 times with concentrates. When compared to the original samples, permeate samples resulted in a lower effect on aggregation lag time ((544 ± 20) minutes, as opposed to (846 ± 28) minutes, based on the data fit in Figure 2A–C). Conversely, concentrate samples caused an increase in the average process lag time values (989 ± 69 min), suggesting that the insoluble or large molecular weight molecules were, on average, more effective at inhibiting aggregation. Despite this higher average value obtained by fitting the data, an ANOVA one-way Bonferroni means comparison between the 21–24 sample sets (n = 16) of all three conditions revealed that there was no statistically significant difference (*p* < 0.01) between the original and concentrate EGCG samples, while they were both significantly different from the permeate samples. Another notable factor was the elevated stochasticity of the fibrillization reaction when in the presence of the concentrate samples, which may stem from the existence of large particles in the solution.

Oppositely to changes in lag time values, the apparent rate constants of aggregation remained generally within the margin of error throughout the entire range of samples (Figure 2D–F), suggesting that the EGCG incubation products did not have a notable effect on insulin aggregate elongation. Finally, it also appeared that the longer incubation time EGCG caused the formation of higher bound-ThT fluorescence intensity samples when compared to the control and the short incubation time samples (Figure 2G–I). ANOVA one-way Bonferroni means comparison of the 1–4 and 21–24 sample sets (n = 16) revealed that there was a significant fluorescence intensity difference (*p* < 0.01) for all three cases. The largest increase in intensity values between the 1–4 and 21–24 sample sets was determined for the concentrate samples (11 times higher average intensity), while the intensity was only 1.6 and 1.7 times higher for the original and permeate samples, respectively. Such fibril formation has been reported previously and attributed to the generation of aggregates with slight structural differences, which facilitate a different mode of ThT binding and cause an increased fluorescence intensity [38]. These large intensity samples only appeared when using 16–24 EGCG samples, which coincided with the appearance of EGCG anti-amyloid properties (Figure 2C,I).

To determine if this effect on lag time is not solely condition-related, the inhibitory effect of EGCG samples was tested under conditions that facilitate a different pathway of insulin aggregation (Appendix B Figure A4). The results revealed a similar tendency, with no effect on lag time observed up to the 16–17 samples, which suggests this emergence of inhibition is not restricted to a certain set of conditions. In addition, it was also examined whether ThT and EGCG did not competitively bind to the aggregates, which could prevent effective inhibition [39]. Insulin fibrils, prepared under AC conditions, were combined with ThT and EGCG in pairs and separately. The fibrils were then pelleted, and the supernatant absorbance spectra were scanned. When ThT and EGCG were combined with fibrils separately, their absorbance spectra were added together in order to compare them to the ThT–EGCG mixture spectra. In both the non-incubated and the incubated EGCG cases, the combined ThT–EGCG mixture yielded lower absorbance spectra than the sum of the separate ThT and EGCG spectra (Appendix B Figure A2D,E). This suggests that not only do the two compounds not interfere when binding to insulin aggregates under such conditions, but they may also experience co-operative binding, as shown previously for different amyloidophilic molecules [40].

In order to evaluate whether the inhibitory effect was the result of stable polymeric particles or low-solubility molecules, the 80 h EGCG sample was subjected to multiple rounds of concentration and subsequent concentrate dilution to its original volume (identical procedure as described previously). Each resulting permeate and the final concentrate were used in an aggregation assay, to determine their respective influence on insulin fibril formation. The first three permeates (Figure 3A) displayed an inhibitory potential similar to the previous analysis (Figure 2B). After four rounds of concentration and dilution, the lag time values began to fall and after seven rounds they were within the margin of error to the control. However, the permeate absorbance spectra (Figure 3B) were significantly reduced after each round of concentration and dilution (~2-fold decrease each time). These results suggest that the initial permeate solution, in addition to inhibitory compounds, also contained a large number of molecules, which were capable of absorbing light in the measured spectrum, but did not have a notable effect on aggregation., We also did not observe an exponential decline in lag time values, which indicates that there were either low-solubility inhibitory molecules in the initial solution (which dissolved and passed through the concentrator after each round of dilution) or there was a fraction of polymeric compounds with lower molecular weights. After the ten rounds of concentration and dilution, the concentrate retained its high inhibitory effect, which was similar to that previously observed (Figure 2C).

As it is known that EGCG can modulate the resulting structure of amyloidogenic protein aggregates, all 24 insulin fibril samples were examined using Fourier-transform infrared spectroscopy (FTIR). All FTIR spectra had the main maximum (minimum in the second derivative) position at 1627 cm^−1^ (Figure 4A,B), suggesting a similar type of dominant hydrogen bonding in the beta-sheet structure [35]. However, there was a discontinuity observed between samples 1–19 (group 1) and 20–24 (group 2). In the case of group 2, the peak at 1627 cm^−1^ became significantly higher (Figure 4C), suggesting a possible increase in beta-sheet content in the fibril structure. There was also a considerable shift in the ratio between 1627 cm^−1^ and 1640 cm^−1^ s derivative minima. This change was similar to previous observations, where different insulin concentrations led to distinct fibril formation [38]. It was also visible when measuring the half-height bandwidth (HHBW) of each sample’s FTIR spectrum (Figure 4D), where the HHBW value experienced a significant reduction past the 19-sample (30 h) mark. These results suggest that after 30 h of incubation, EGCG gains the property of altering the secondary structure of the resulting insulin fibril aggregates. These induced differences are mainly related to the quantity of structural motifs present in the fibril (significantly different peak areas), rather than their type (similar peak positions).

The two group fibrils were also examined using atomic force microscopy (AFM). In both cases, insulin formed long and straight fibrils (up to several micrometers in length) with no discernable periodicity patterns (Figure 4E,F). When comparing the average cross-sectional height of the aggregates of samples 1 and 24, it was observed that this parameter was significantly higher (n = 30, *p* < 0.01) when insulin was aggregated without EGCG (Figure 4G). In the case of fibril width, there were no significant differences between the two samples (Figure 4H).

Since it has been reported that EGCG can cause insulin precipitation, which may factor into the observed inhibitory effect, the interaction between protein and inhibitor molecules was observed by scanning sample optical densities at 800 nm (OD_800_). When non-aggregated insulin was combined with non-incubated EGCG, there were no notable changes in sample OD_800_ values (Figure 5A), even when the concentration of EGCG was increased to 400 µM (16 times higher than used in the aggregation assay). Matters became significantly different when the same examination was carried out with the 80-h incubation sample (Figure 5A). There appeared to be an almost linear dependence between the concentration of added EGCG and the sample optical density, indicating that molecules present in the inhibitor sample cause concentration-dependent precipitation of insulin. Without insulin, the same concentration of EGCG (400 µM) resulted in an OD value within the margin of the control, and only the 10 mM (25 times higher concentration) original 80-h incubation sample had a comparable optical density (Appendix B, Figure A2B).

To determine which of the EGCG samples had the capability of precipitating insulin, a similar experiment was carried out with all 24 samples. Based on sample OD_800_ values, it appeared that only 20–24 samples were capable of inducing a significant level of insulin precipitation, with all other samples being within the margin of error to the control (Figure 5B). Interestingly, the 50 µM EGCG sample did not show a significant difference from the control (Figure 5A) and the appearance of precipitates with only 20–24 samples (Figure 5B) suggested that this factor may not be detrimental to the previously observed inhibitory effect and only played a part in the process.

## 4. Discussion

Taking into consideration the gradual changes in EGCG absorbance spectra (Figure 1A,B), formation of precipitates, and larger molecular weight molecules (Figure 1D,E) that can be separated using concentrators, as well as previous reports [28,41] of multiple possible EGCG aggregation products, it is quite clear that this is a highly complex, multi-stage process, involving an array of different structural transitions. The fact that considerable changes to this inhibitor molecule begin to occur after as little as 1 h of incubation at an elevated temperature calls into question the results and conclusions obtained from studies using EGCG, especially when the examinations are carried out under conditions that facilitate its autoxidation. In order to accurately determine what effect both the initial molecule as well as its autoxidation products have on insulin amyloid aggregation, all procedures involved in this work were carried out under conditions that minimize its autoxidation, from initial sample preparation at low temperatures to aggregation reactions under acidic pH.

The aggregation assay revealed that non-incubated EGCG had no notable effect on insulin amyloid aggregation under both 20% acetic acid conditions, where insulin is present in its monomeric state, as well as pH 2.4 conditions, where insulin is dimeric. In fact, the anti-amyloid effect only became visible after 15 h of EGCG incubation at 60 °C, with the effect reaching a plateau after 40 h of incubation. A similar time frame was needed for the inhibitor to lead to the formation of high bound-ThT fluorescence emission samples, where the fibrils have slightly different surface characteristics that can facilitate distinct dye binding modes [38]. Clear distinctions in fibril FTIR spectra only became visible after EGCG had been incubated for 40 h or more. Since these structural changes occur alongside a considerable increase in lag time, the appearance of higher bound-ThT fluorescence intensity samples, as well as with EGCG gaining its property to precipitate insulin, it is difficult to point toward one factor that causes the structural transition.

Another interesting aspect is the divide between the inhibitory effect of molecules that were able to pass through a 10 kDa concentrator and the retentate/concentrate. The permeate was considerably weaker at affecting insulin aggregation lag time, while the concentrate was, on average, more efficient than the original samples, suggesting that the larger molecular weight or lower solubility compounds are some of the most potent inhibitors present in the EGCG product mixture. Further support for this idea was gained from the procedure of multiple EGCG dilution and concentration steps, where the permeate gradually lost its inhibitory potential, while the concentrate remained a highly effective inhibitor. If this is the case, then the low solubility and/or large size creates a potential problem for the future prospects of applying these compounds in vivo, as well as complicating their identification and characterization procedures.

Regarding reports of seemingly non-oxidized EGCG having a prominent effect on insulin or other protein/peptide aggregation, there can be a few possible explanations. In some cases, protein aggregation occurs over a span of several hours or days at physiological or higher temperatures and neutral pH. This makes it possible for EGCG to form autoxidation products with anti-amyloid properties during the primary nuclei formation phase. In other cases, the preparation and storage conditions of EGCG can lead to the gradual formation of such compounds, as even room temperature conditions cause a slow autoxidation process. This is best exemplified in a study by Zhang et al., where even a short incubation period under physiological conditions resulted in the formation of theasinensin A, theasinensin D, a dimer quinone, and other polyphenolic compounds [28]. Other studies have also shown that during autoxidation, EGCG can undergo stereochemical rearrangements, as well as oligomerization and polymerization steps [42,43]. If certain types of these intermediate species are responsible for the aggregation-inhibiting effect, then conditions different from the ones used in this work may cause their appearance after a shorter timespan or even generate distinct, anti-amyloid compounds.

This work shows that all previously reported effects associated with EGCG, such as increased aggregation lag time, changes to fluorescence intensity, secondary structure, morphology, and even protein precipitation are, in fact, correct. However, the observed anti-amyloid activity may not be related to the EGCG molecule in question, but rather to EGCG autoxidation products, which require a certain specific time period to form. The results obtained here display the complex mode of EGCG autoxidation, as well as the possibility of the autoxidation products being composed of both a wide array of small molecular weight compounds, as well as polymeric species and low solubility molecules.

## Figures and Tables

**Figure 1 antioxidants-11-01887-f001:**
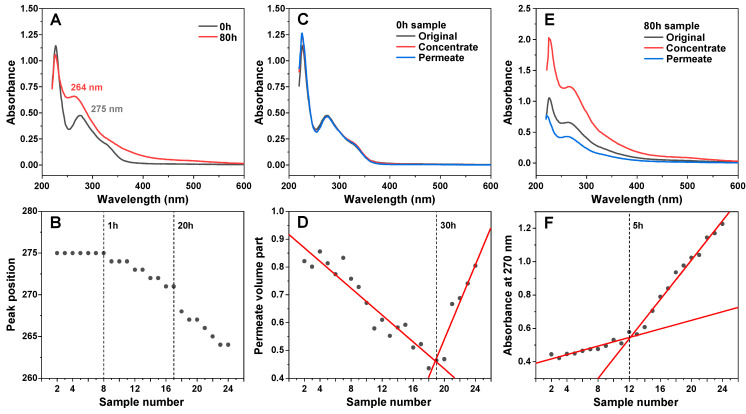
EGCG sample absorbance and membrane permeability changes during incubation. Absorbance spectra of EGCG before and after 80 h of incubation at 60 °C (**A**) and a shift in its characteristic peak position during the incubation time frame (**B**). Non-incubated EGCG sample concentrate and permeate absorbance spectra (**C**). Sample permeate volume part dependence on sample number (**D**). EGCG sample concentrate and permeate absorbance spectra after 80 h incubation (**E**). Concentrate absorbance at 270 nm by on sample number (**F**). Sample preparation and measurement procedures are described in the Section 2.

**Figure 2 antioxidants-11-01887-f002:**
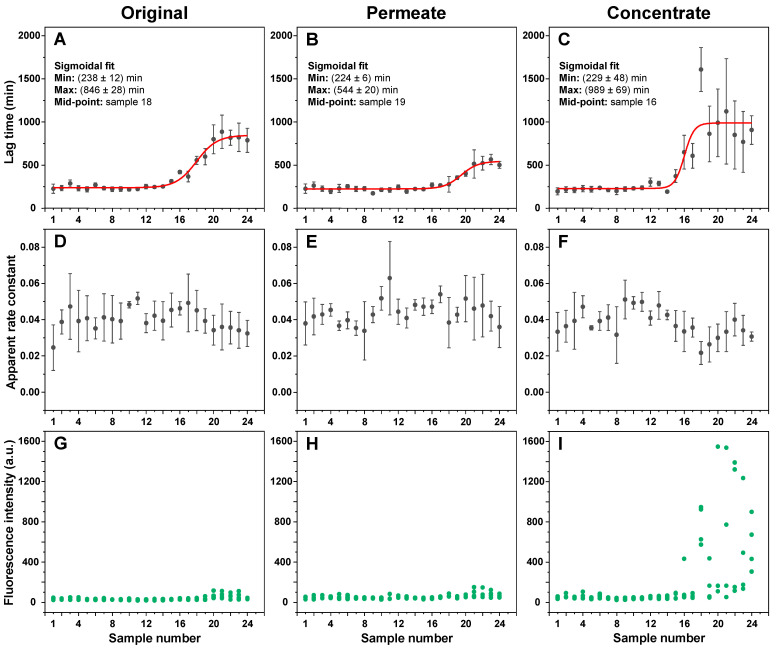
Insulin aggregation kinetic parameters in the presence of different incubation time EGCG samples. Insulin aggregation lag times when in the presence of original (**A**), permeate (**B**), and concentrate (**C**) EGCG samples, their elongation apparent rate constants (**D**–**F** respectively), and sample bound-ThT fluorescence intensities at the end of the reaction (**G**–**I** respectively). EGCG sample preparation is described in the Section 2. Aggregation lag time, apparent rate constant, and sample fluorescence intensity values were determined as shown in Appendix B, Figure A1 (n = 4, error bars are for one standard deviation). Examples for aggregation kinetic curves of samples 1 and 24 are shown in Appendix B, Figure A3. Inserts in subfigures (**A**–**C**) show the minimum and maximum lag time values and mid-point of the sigmoidal fit.

**Figure 3 antioxidants-11-01887-f003:**
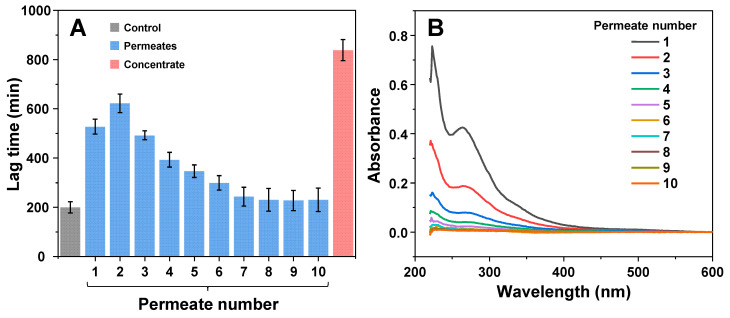
Effect of EGCG solution permeates and remaining concentrate on insulin aggregation after multiple rounds of concentration and dilution. The aggregation kinetics (**A**) and permeate absorbance spectra (**B**) scans were performed as described in the Section 2. Lag time values were determined from four repeats (error bars are for one standard deviation). Absorbance spectra are the average of three repeats.

**Figure 4 antioxidants-11-01887-f004:**
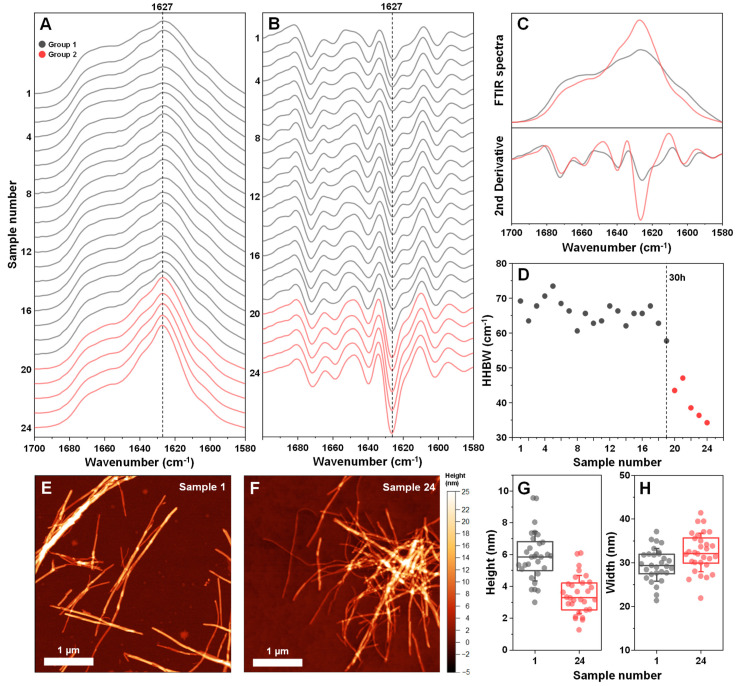
Fourier-transform infrared (FTIR) spectra and atomic force microscopy (AFM) images of insulin fibrils, generated using different incubation time EGCG samples. Twenty-four different insulin fibril FTIR spectra (**A**) and their second derivatives (**B**), prepared using a range of different incubation time EGCG samples. Samples are divided into two groups (group 1–grey color, group 2–red color) based on differences in their FTIR spectra. Comparison of both group FTIR spectra and their second derivatives (**C**). Half-height bandwidth (HHBW) of all 24 FTIR spectra (**D**). AFM images of sample 1 (**E**) and sample 24 (**F**) insulin fibrils and their respective cross-sectional height (**G**) and width (**H**) distribution (n = 30). FTIR spectra and AFM image acquisition is described in the Section 2. Different group spectra and data points are color-coded in all subfigures.

**Figure 5 antioxidants-11-01887-f005:**
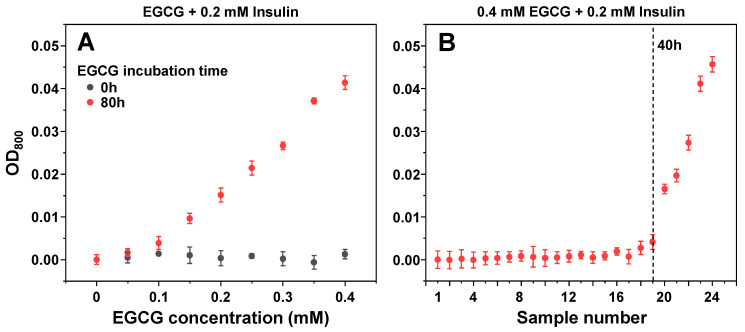
Optical density of insulin samples in the presence of different concentration (**A**) and incubation time (**B**) EGCG samples. Optical density measurement procedures are described in the Section 2. Optical density values are the average of three repeats (error bars are for one standard deviation).

**Table 1 antioxidants-11-01887-t001:** EGCG sample incubation times and their numbering. Sample 1 did not contain EGCG and was used as a control, samples 2–24 contained 10 mM EGCG.

Sample Number	Incubation Time	Sample Number	Incubation Time	Sample Number	Incubation Time
1	No EGCG	9	2 h	17	20 h
2	0 min	10	3 h	18	25 h
3	10 min	11	4 h	19	30 h
4	20 min	12	5 h	20	40 h
5	30 min	13	6 h	21	50 h
6	40 min	14	9 h	22	60 h
7	50 min	15	12 h	23	70 h
8	1 h	16	15 h	24	80 h

## Data Availability

The data presented in this study are available in Appendix A.

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
