# Peer review of "Exploring Epigallocatechin-3-Gallate Autoxidation Products: Specific Incubation Times Required for Emergence of Anti-Amyloid Properties"

_antioxidants, 2022, doi:10.3390/antiox11101887_

Round 1

Reviewer 1 Report

The word "perfect" should be removed from the introduction.

Reviewer 2 Report

This study investigated the oxidation process of EGCG and the its effect on the inhibition of insulin amyloid aggregation; EGCG oxide was fractionated by filtration device according to incubation time, and the UV-visible spectrum was shown to change. Furthermore, the dependence of the insulin aggregation inhibitory effect on EGCG incubation time was shown by ThT fluorescence intensity time course measurements and FTIR. I recommend that the followings should be verified prior to publication.

  1. Polyphenols, including EGCG, are in a competitive inhibitory relationship with ThT in binding to amyloid aggregates in ThT measurements (https://doi.org/10.1177%2F1934578X19849791https://doi.org/10.1016/j.bpc.2017.07.009https://doi.org/10.1021/acsami.6b06853).
    Does this study confirm this competitive inhibition
    ? Could the degree of competitive inhibition vary depending on the incubation time of EGCG? It should be verified if the same conclusion can be made using other methods (Congo red, TEM, etc.).
  2. This study only shows spectroscopic data (ThT, FTIR) for amyloid aggregation and does not show morphology data. The fluorescence intensity of ThT can be incleased even if the amyloid fibrils are not main. Nevertheless, the word "fibril" is often used in the manuscript; morphology should be shown by TEM, AFM, etc.

Reviewer 3 Report

This study investigates an interesting area of identifying the specific form of EGCG that may be responsible for its reported aggregation inhibition activity. The overall data is largely well presented but I found the section comparing the separated samples difficult to read. While the summary graphs of lag time, rate etc are helpful I would like to see the inclusion of raw ThT curves in the appendix. I also have a few comments regarding the authors use of language when talking about differences observed:

How did the authors assess significance (and what were the significance values generated) for the statement on line 210 comparing original and permeate samples on aggregation lag time? 

On line 232 the authors describe the data as considerably more intense than the small increase observed for both the original and permeate samples, was any statistical analysis done on this data? If not, could it be done?

Can the authors provide more details on the conditions and alternative aggregation pathway data presented in Figure A3.

When describing generation of multiple permeates the authors describe the first 3 permeates as not having a notable effect on aggregation. It is not clear what this is in comparison to. This should be clarified - as there is a difference compared to insulin aggregation alone but not original incubated EGCG sample.

Figure 4 legend should contain a description of groups 1 and 2.

Line 303 -  10 mM (25 times higher concentration) seems to be incorrect.

The discussion would benefit from additional information about what is known about EGCG oxidation at neutral pH and lower temperature e.g. 37oC to make this research more widely applicable to other aggregation studies.

Round 2

Reviewer 2 Report

The revised manuscript extensively addresses the Reviewer's comments. It is recommended that the color scale of the AFM images in Figures 4E and 4F (the correspondence between the height of the sample from the stage and its color) be added prior to publication.
